# TRAINING FOR FASTER ADVERSARIAL ROBUSTNESS VERIFICATION VIA INDUCING ReLU STABILITY

**Kai Y. Xiao    Vincent Tjeng    Nur Muhammad (Mahi) Shafiullah    Aleksander Mądry**
Massachusetts Institute of Technology
Cambridge, MA 02139
{kaix, vtjeng, nshafiul, madry}@mit.edu

## ABSTRACT

We explore the concept of co-design in the context of neural network verification. Specifically, we aim to train deep neural networks that not only are robust to adversarial perturbations but also whose robustness can be verified more easily. To this end, we identify two properties of network models – weight sparsity and so-called ReLU stability – that turn out to significantly impact the complexity of the corresponding verification task. We demonstrate that improving weight sparsity alone already enables us to turn computationally intractable verification problems into tractable ones. Then, improving ReLU stability leads to an additional 4–13x speedup in verification times. An important feature of our methodology is its "universality," in the sense that it can be used with a broad range of training procedures and verification approaches.

## 1 INTRODUCTION

Deep neural networks (DNNs) have recently achieved widespread success in image classification (Krizhevsky et al., 2012), face and speech recognition (Taigman et al., 2014; Hinton et al., 2012), and game playing (Silver et al., 2016; 2017). This success motivates their application in a broader set of domains, including more safety-critical environments. This thrust makes understanding the reliability and robustness of the underlying models, let alone their resilience to manipulation by malicious actors, a central question. However, predictions made by machine learning models are often brittle. A prominent example is the existence of adversarial examples (Szegedy et al., 2014): imperceptibly modified inputs that cause state-of-the-art models to misclassify with high confidence.

There has been a long line of work on both generating adversarial examples, called *attacks* (Carlini and Wagner, 2017a;b; Athalye et al., 2018a;b; Uesato et al., 2018; Evtimov et al., 2017), and training models robust to adversarial examples, called *defenses* (Goodfellow et al., 2015; Papernot et al., 2016; Madry et al., 2018; Kannan et al., 2018). However, recent research has shown that most defenses are ineffective (Carlini and Wagner, 2017a; Athalye et al., 2018a; Uesato et al., 2018). Furthermore, even for defenses such as that of Madry et al. (2018) that have seen empirical success against many attacks, we are unable to conclude yet with certainty that they are robust to all attacks that we want these models to be resilient to.

This state of affairs gives rise to the need for *verification of networks*, i.e., the task of *formally* proving that no small perturbations of a given input can cause it to be misclassified by the network model. Although many exact verifiers[1] have been designed to solve this problem, the verification process is often intractably slow. For example, when using the Reluplex verifier of Katz et al. (2017), even verifying a small MNIST network turns out to be computationally infeasible. Thus, addressing this intractability of exact verification is the primary goal of this work.

**Our Contributions**
Our starting point is the observation that, typically, model training and verification are decoupled and seen as two distinct steps. Even though this separation is natural, it misses a key opportunity: the ability to align these two stages. Specifically, applying the principle of *co-design* during model

---

[1]Also sometimes referred to as combinatorial verifiers.

training is possible: training models in a way to encourage them to be simultaneously robust and easy-to-exactly-verify. This insight is the cornerstone of the techniques we develop in this paper.

In this work, we use the principle of co-design to develop training techniques that give models that are both robust and easy-to-verify. Our techniques rely on improving two key properties of networks: weight sparsity and ReLU stability. Specifically, we first show that natural methods for improving weight sparsity during training, such as $\ell_1$-regularization, give models that can already be verified much faster than current methods. This speedup happens because in general, exact verifiers benefit from having fewer variables in their formulations of the verification task. For instance, for exact verifiers that rely on linear programming (LP) solvers, sparser weight matrices means fewer variables in those constraints.

We then focus on the major speed bottleneck of current approaches to exact verification of ReLU networks: the need of exact verification methods to "branch," i.e., consider two possible cases for each ReLU (ReLU being active or inactive). Branching drastically increases the complexity of verification. Thus, well-optimized verifiers will not need to branch on a ReLU if it can determine that the ReLU is *stable*, i.e. that the ReLU will always be active or always be inactive for any perturbation of an input. This motivates the key goal of the techniques presented in this paper: we aim to minimize branching by maximizing the number of stable ReLUs. We call this goal *ReLU stability* and introduce a regularization technique to induce it.

Our techniques enable us to train weight-sparse and ReLU stable networks for MNIST and CIFAR-10 that can be verified significantly faster. Specifically, by combining natural methods for inducing weight sparsity with a robust adversarial training procedure (cf. Goodfellow et al. (2015)), we are able to train networks for which almost $90\%$ of inputs can be verified in an amount of time that is small[2] compared to previous verification techniques. Then, by also adding our regularization technique for inducing ReLU stability, we are able to train models that can be verified an additional 4–13x times as fast while maintaining state-of-the-art accuracy on MNIST. Our techniques show similar improvements for exact verification of CIFAR models. In particular, we achieve the following verification speed and provable robustness results for $\ell_\infty$ norm-bound adversaries:

| Dataset | Epsilon | Provable Adversarial Accuracy | Average Solve Time (s) |
|---------|---------|:-----------------------------:|:----------------------:|
| MNIST | $\epsilon = 0.1$ | 94.33% | 0.49 |
|  | $\epsilon = 0.2$ | 89.79% | 1.13 |
|  | $\epsilon = 0.3$ | 80.68% | 2.78 |
| CIFAR | $\epsilon = 2/255$ | 45.93% | 13.50 |
|  | $\epsilon = 8/255$ | 20.27% | 22.33 |

Our network for $\epsilon = 0.1$ on MNIST achieves provable adversarial accuracies comparable with the current best results of Wong et al. (2018) and Dvijotham et al. (2018), and our results for $\epsilon = 0.2$ and $\epsilon = 0.3$ achieve the best provable adversarial accuracies yet. To the best of our knowledge, we also achieve the first nontrivial provable adversarial accuracy results using exact verifiers for CIFAR-10.

Finally, we design our training techniques with universality as a goal. We focus on improving the *input* to the verification process, regardless of the verifier we end up using. This is particularly important because research into network verification methods is still in its early stages, and our co-design methods are compatible with the best current verifiers (LP/MILP-based methods) and should be compatible with any future improvements in verification.

Our code is available at `https://github.com/MadryLab/relu_stable`.

## 2   BACKGROUND AND RELATED WORK

Exact verification of networks has been the subject of many recent works (Katz et al., 2017; Ehlers, 2017; Carlini et al., 2017; Tjeng et al., 2017; Lomuscio and Maganti, 2017; Cheng et al., 2017a). To understand the context of these works, observe that for linear networks, the task of exact verification is relatively simple and can be done by solving a LP. For more complex models, the presence of nonlinear ReLUs makes verification over all perturbations of an input much more challenging. This

---

[2]We chose our time budget for verification to be 120 seconds per input image.

is so as ReLUs can be active or inactive depending on the input, which can cause exact verifiers to "branch" and consider these two cases separately. The number of such cases that verifiers have to consider might grow exponentially with the number of ReLUs, so verification speed will also grow exponentially in the worst case. Katz et al. (2017) further illustrated the difficulty of exact verification by proving that it is NP-complete. In recent years, formal verification methods were developed to verify networks. Most of these methods use satisfiability modulo theory (SMT) solvers (Katz et al., 2017; Ehlers, 2017; Carlini et al., 2017) or LP and Mixed-Integer Linear Programming (MILP) solvers (Tjeng et al., 2017; Lomuscio and Maganti, 2017; Cheng et al., 2017a). However, all of them are limited by the same issue of scaling poorly with the number of ReLUs in a network, making them prohibitively slow in practice for even medium-sized models.

One recent approach for dealing with the inefficiency of exact verifiers is to focus on certification methods[3] (Wong and Kolter, 2018; Wong et al., 2018; Dvijotham et al., 2018; Raghunathan et al., 2018; Mirman et al., 2018; Sinha et al., 2018). In contrast to exact verification, these methods do not solve the verification task directly; instead, they rely on solving a *relaxation* of the verification problem. This relaxation is usually derived by overapproximating the adversarial polytope, or the space of outputs of a network for a region of possible inputs. These approaches rely on training models in a specific manner that makes certification of those models easier. As a result, they can often obtain provable adversarial accuracy results faster. However, certification is fundamentally different from verification in two primary ways. First, it solves a relaxation of the original verification problem. As a result, certification methods can fail to certify many inputs that are actually robust to perturbations – only exact verifiers, given enough time, can give conclusive answers on robustness for every single input. Second, certification approaches fall under the paradigm of co-training, where a certification method only works well on models specifically trained for that certification step. When used as a black box on arbitrary models, the certification step can yield a high rate of false negatives. For example, Raghunathan et al. (2018) found that their certification step was significantly less effective when used on a model trained using Wong and Kolter (2018)'s training method, and vice versa. In contrast, we design our methods to be universal. They can be combined with any standard training procedure for networks and will improve exact verification speed for any LP/MILP-based exact verifier. Our methods can also decrease the amount of overapproximation incurred by certification methods like Wong and Kolter (2018); Dvijotham et al. (2018). Similar to most of the certification methods, our technique can be made to have very little training time overhead.

Finally, subsequent work of Gowal et al. (2018) shows how applying interval bound propagation during training, combined with MILP-based exact verification, can lead to provably robust networks.

## 3 TRAINING VERIFIABLE NETWORK MODELS

We begin by discussing the task of verifying a network and identify two key properties of networks that lead to improved verification speed: weight sparsity and so-called ReLU stability. We then use natural regularization methods for inducing weight sparsity as well as a new regularization method for inducing ReLU stability. Finally, we demonstrate that these methods significantly speed up verification while maintaining state-of-the-art accuracy.

### 3.1 VERIFYING ADVERSARIAL ROBUSTNESS OF NETWORK MODELS

**Deep neural network models.** Our focus will be on one of the most common architectures for state-of-the-art models: $k$-layer fully-connected feed-forward DNN classifiers with ReLU non-linearities[4]. Such models can be viewed as a function $f(\cdot, W, b)$, where $W$ and $b$ represent the weight matrices and biases of each layer. For an input $x$, the output $f(x, W, b)$ of the DNN is

---

[3]These works use both "verification" and "certification" to describe their methods. For clarity, we use "certification" to describe their approaches, while we use "verification" to describe *exact* verification approaches. For a more detailed discussion of the differences, see Appendix F.

[4]Note that this encompasses common convolutional network architectures because every convolutional layer can be replaced by an equivalent fully-connected layer.

defined as:

$$z_0 = x \tag{1}$$
$$\hat{z}_i = z_{i-1} W_i + b_i \quad \text{for } i = 1, 2, \ldots, k-1 \tag{2}$$
$$z_i = \max(\hat{z}_i, 0) \quad \text{for } i = 1, 2, \ldots, k-2 \tag{3}$$
$$f(x, W, b) = \hat{z}_{k-1} \tag{4}$$

Here, for each layer $i$, we let $\hat{z}_{ij}$ denote the $j^{th}$ ReLU pre-activation and let $\hat{z}_{ij}(x)$ denote the value of $\hat{z}_{ij}$ on an input $x$. $\hat{z}_{k-1}$ is the final, output layer with an output unit for each possible label (the logits). The network will make predictions by selecting the label with the largest logit.

**Adversarial robustness.** For a network to be reliable, it should make predictions that are robust – that is, it should predict the same output for inputs that are very similar. Specifically, we want the DNN classifier's predictions to be robust to a set $\text{Adv}(x)$ of possible adversarial perturbations of an input $x$. We focus on $\ell_\infty$ norm-bound adversarial perturbations, where $\text{Adv}(x) = \{x' : ||x' - x||_\infty \leq \epsilon\}$ for some constant $\epsilon$, since it is the most common one considered in adversarial robustness and verification literature (thus, it currently constitutes a canonical benchmark). Even so, our methods can be applied to other $\ell_p$ norms and broader sets of perturbations.

**Verification of network models.** For an input $x$ with correct label $y$, a perturbed input $x'$ can cause a misclassification if it makes the logit of some incorrect label $\hat{y}$ larger than the logit of $y$ on $x'$. We can thus express the task of finding an adversarial perturbation as the optimization problem:

$$\min_{x', \hat{y}} f(x', W)_y - f(x', W)_{\hat{y}}$$
$$\text{subject to} \quad x' \in \text{Adv}(x)$$

An adversarial perturbation exists if and only if the objective of the optimization problem is negative.

**Adversarial accuracies.** We define the *true adversarial accuracy* of a model to be the fraction of test set inputs for which the model is robust to all allowed perturbations. By definition, evaluations against specific adversarial attacks like PGD or FGSM provide an upper bound to this accuracy, while certification methods provide lower bounds. Given sufficient time for each input, an exact verifier can prove robustness to perturbations, or find a perturbation where the network makes a misclassification on the input, and thus exactly determine the true adversarial accuracy. This is in contrast to certification methods, which solve a relaxation of the verification problem and can not exactly determine the true adversarial accuracy no matter how much time they have.

However, such exact verification may take impractically long for certain inputs, so we instead compute the *provable adversarial accuracy*, which we define as the fraction of test set inputs for which the verifier can prove robustness to perturbations within an allocated time budget (timeout). Similarly to certifiable accuracy, this accuracy constitutes a lower bound on the true adversarial accuracy. A model can thus, e.g., have high true adversarial accuracy and low provable adversarial accuracy if verification of the model is too slow and often fails to complete before timeout.

Also, in our evaluations, we chose to use the MILP exact verifier of Tjeng et al. (2017) when performing experiments, as it is both open source and the fastest verifier we are aware of.

## 3.2 Weight Sparsity and its Impact on Verification Speed

The first property of network models that we want to improve in order to speed up exact verification of those models is weight sparsity. Weight sparsity is important for verification speed because many exact verifiers rely on solving LP or MILP systems, which benefit from having fewer variables. We use two natural regularization methods for improving weight sparsity: $\ell_1$-regularization and small weight pruning. These techniques significantly improve verification speed – see Table 1. Verifying even small MNIST networks is almost completely intractable without them. Specifically, the verifier can only prove robustness of an adversarially-trained model on $19\%$ of inputs with a one hour budget per input, while the verifier can prove robustness of an adversarially-trained model with $\ell_1$-regularization and small weight pruning on $89.13\%$ of inputs with a 120 second budget per input.

Interestingly, even though adversarial training improves weight sparsity (see Appendix B) it was still necessary to use $\ell_1$-regularization and small weight pruning. These techniques further promoted weight sparsity and gave rise to networks that were much easier to verify.

| Dataset | Epsilon | | Training Method | Test Set Accuracy | Provable Adversarial Accuracy | Average Solve Time (s) |
|---|---|---|---|---|---|---|
| MNIST | $\epsilon = 0.1$ | 1 | Adversarial Training | 99.17% | 19.00% | 2970.43 |
| | | 2 | $+\ell_1$-Regularization | 99.00% | 82.17% | 21.99 |
| | | 3 | +Small Weight Pruning | 98.99% | 89.13% | 11.71 |
| | | 4 | +ReLU Pruning (control) | 98.94% | 91.58% | 6.43 |

Table 1: Improvement in provable adversarial accuracy and verification solve times when incrementally adding natural regularization methods for improving weight sparsity and ReLU stability into the model training procedure, before verification occurs. Each row represents the addition of another method – for example, Row 3 uses adversarial training, $\ell_1$-regularization, and small weight pruning. Row 4 adds ReLU pruning (see Appendix A). Row 4 is the control model for MNIST and $\epsilon = 0.1$ that we present again in comparisons in Tables 2 and 3. We use a 3600 instead of 120 second timeout for Row 1 and only verified the first 100 images (out of 10000) because verifying it took too long.

## 3.3 ReLU Stability

Next, we target the primary speed bottleneck of exact verification: the number of ReLUs the verifier has to branch on. In our paper, this corresponds to the notion of inducing ReLU stability. Before we describe our methodology, we formally define ReLU stability.

Given an input $x$, a set of allowed perturbations $\text{Adv}(x)$, and a ReLU, exact verifiers may need to branch based on the possible pre-activations of the ReLU, namely $\hat{z}_{ij}(\text{Adv}(x)) = \{\hat{z}_{ij}(x') : x' \in \text{Adv}(x)\}$ (cf. (2)). If there exist two perturbations $x', x''$ in the set $\text{Adv}(x)$ such that $\text{sign}(\hat{z}_{ij}(x')) \neq \text{sign}(\hat{z}_{ij}(x''))$, then the verifier has to consider that for some perturbed inputs the ReLU is active ($z_{ij} = \hat{z}_{ij}$) and for other perturbed inputs inactive ($z_{ij} = 0$). The more such cases the verifier faces, the more branches it has to consider, causing the complexity of verification to increase exponentially. Intuitively, a model with 1000 ReLUs among which only 100 ReLUs require branching will likely be much easier to verify than a model with 200 ReLUs that all require branching. Thus, it is advantageous for the verifier if, on an input $x$ with allowed perturbation set $\text{Adv}(x)$, the number of ReLUs such that

$$\text{sign}(\hat{z}_{ij}(x')) = \text{sign}(\hat{z}_{ij}(x)) \quad \forall x' \in \text{Adv}(x) \tag{5}$$

is maximized. We call a ReLU for which (5) holds on an input $x$ a *stable ReLU* on that input. If (5) does not hold, then the ReLU is an *unstable ReLU*.

Directly computing whether a ReLU is stable on a given input $x$ is difficult because doing so would involve considering all possible values of $\hat{z}_{ij}(\text{Adv}(x))$. Instead, exact verifiers compute an upper bound $\hat{u}_{ij}$ and a lower bound $\hat{l}_{ij}$ of $\hat{z}_{ij}(\text{Adv}(x))$. If $0 \leq \hat{l}_{ij}$ or $\hat{u}_{ij} \leq 0$, they can replace the ReLU with the identity function or the zero function, respectively. Otherwise, if $\hat{l}_{ij} < 0 < \hat{u}_{ij}$, these verifiers then determine that they need to "branch" on that ReLU. Thus, we can rephrase (5) as

$$\text{sign}(\hat{u}_{ij}) = \text{sign}(\hat{l}_{ij}) \tag{6}$$

We will discuss methods for determining these upper and lower bounds $\hat{u}_{ij}, \hat{l}_{ij}$ in Section 3.3.2.

### 3.3.1 A Regularization Technique for Inducing ReLU Stability: RS Loss

As we see from equation (6), a function that would indicate exactly when a ReLU is stable is $F^*(\hat{u}_{ij}, \hat{l}_{ij}) = \text{sign}(\hat{u}_{ij}) \cdot \text{sign}(\hat{l}_{ij})$. Thus, it would be natural to use this function as a regularizer. However, this function is non-differentiable and if used in training a model, would provide no useful gradients during back-propagation. Thus, we use the following smooth approximation of $F^*$ (see Fig. 1) which provides the desired gradients:

$$F(\hat{u}_{ij}, \hat{l}_{ij}) = -\tanh(1 + \hat{u}_{ij} \cdot \hat{l}_{ij})$$

We call the corresponding objective RS Loss, and show in Fig. 2a that using this loss function as a regularizer effectively decreases the number of unstable ReLUs. RS Loss thus encourages *ReLU stability*, which, in turn, speeds up exact verification - see Fig. 2b.

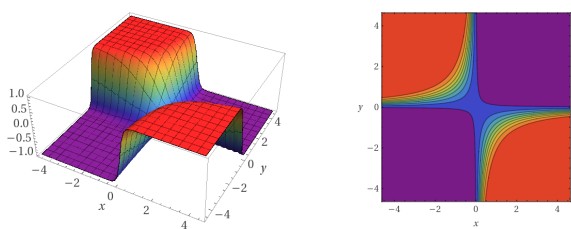

Figure 1: Plot and contour plot of the function $F(x, y) = -\tanh(1 + x \cdot y)$

### 3.3.2 Estimating ReLU Upper and Lower Bounds on Activations

A key aspect of using RS Loss is determining the upper and lower bounds $\hat{u}_{ij}, \hat{l}_{ij}$ for each ReLU (cf. (6)). The bounds for the inputs $z_0$ (cf. (1)) are simple – for the input $x$, we know $x - \epsilon \le z_0 \le x + \epsilon$, so $\hat{u}_0 = x - \epsilon, \hat{l}_0 = x + \epsilon$. For all subsequent $z_{ij}$, we estimate bounds using either the naive interval arithmetic (IA) approach described in Tjeng et al. (2017) or an improved version of it. The improved version is a tighter estimate but uses more memory and training time, and thus is most effective on smaller networks. We present the details of naive IA and improved IA in Appendix C.

Interval arithmetic approaches can be implemented relatively efficiently and work well with back-propagation because they only involve matrix multiplications. This contrasts with how exact verifiers compute these bounds, which usually involves solving LPs or MILPs. Interval arithmetic also overestimates the number of unstable ReLUs. This means that minimizing unstable ReLUs based on IA bounds will provide an upper bound on the number of unstable ReLUs determined by exact verifiers. In particular, IA will properly penalize every unstable ReLU.

Improved IA performs well in practice, overestimating the number of unstable ReLUs by less than $0.4\%$ in the first 2 layers of MNIST models and by less than $36.8\%$ (compared to $128.5\%$ for naive IA) in the 3rd layer. Full experimental results are available in Table 4 of Appendix C.3.

### 3.3.3 Impact of ReLU Stability Improvements on Verification Speed

We provide experimental evidence that RS Loss regularization improves ReLU stability and speeds up average verification times by more than an order of magnitude in Fig. 2b. To isolate the effect of RS Loss, we compare MNIST models trained in exactly the same way other than the weight on RS Loss. When comparing a network trained with a RS Loss weight of $5e-4$ to a network with a RS Loss weight of 0, the former has just $16\%$ as many unstable ReLUs and can be verified 65x faster. The caveat here is that the former has $1.00\%$ lower test set accuracy.

We also compare verification speed with and without RS Loss on MNIST networks for different values of $\epsilon$ (0.1, 0.2, and 0.3) in Fig. 2c. We choose RS Loss weights that cause almost no test set accuracy loss (less than $0.50\%$ - See Table 3) in these cases, and we still observe a 4–13x speedup from RS Loss. For CIFAR, RS Loss gives a smaller speedup of 1.6–3.7x (See Appendix E).

### 3.3.4 Impact of ReLU Stability Improvements on Provable Adversarial Accuracy

As the weight on the RS Loss used in training a model increases, the ReLU stability of the model will improve, speeding up verification and likely improving provable adversarial accuracy. However, like most regularization methods, placing too much weight on RS Loss can decrease the model capacity, potentially lowering both the true adversarial accuracy and the provable adversarial accuracy. Therefore, it is important to choose the weight on RS Loss carefully to obtain both high provable adversarial accuracy and faster verification speeds.

To show the effectiveness of RS Loss in improving provable adversarial accuracy, we train two networks for each dataset and each value of $\epsilon$. One is a "*control*" network that uses all of the natural improvements for inducing both weight sparsity ($\ell_1$-regularization and small weight pruning) and ReLU stability (ReLU pruning - see Appendix A). The second is a "*+RS*" network that uses RS

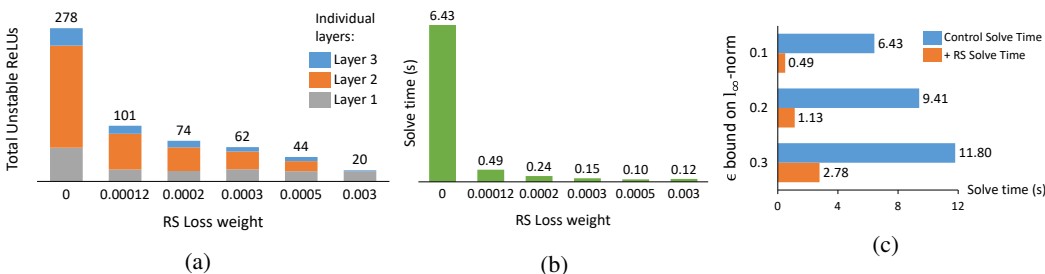

Figure 2: (a) Average number of unstable ReLUs by layer and (b) average verification solve times of 6 networks trained with different weights on RS Loss for MNIST and $\epsilon = 0.1$ . Averages are taken over all 10000 MNIST test set inputs. Both metrics improve significantly with increasing RS Loss weight. An RS Loss weight of 0 corresponds to the control network, while an RS Loss weight of 0.00012 corresponds to the "+RS" network for MNIST and $\epsilon = 0.1$ in Tables 2 and 3. (c) Improvement in the average time taken by a verifier to solve the verification problem after adding RS Loss to the training procedure, for different $\epsilon$ on MNIST. The weight on RS Loss was chosen so that the "+RS" models have test set accuracies within $0.50\%$ of the control models.

Loss in addition to all of the same natural improvements. This lets us isolate the incremental effect of adding RS Loss to the training procedure.

In addition to attaining a 4–13x speedup in MNIST verification times (see Fig. 2c), we achieve higher provable adversarial accuracy in every single setting when using RS Loss. This is especially notable for the hardest verification problem we tackle – proving robustness to perturbations with $\ell_\infty$ norm-bound $8/255$ on CIFAR-10 – where adding RS Loss nearly triples the provable adversarial accuracy from $7.09\%$ to $20.27\%$. This improvement is primarily due to verification speedup induced by RS Loss, which allows the verifier to finish proving robustness for far more inputs within the 120 second time limit. These results are shown in Table 2.

Table 2: Provable Adversarial Accuracies for the control and "+RS" networks in each setting.

|  | MNIST, $\epsilon = 0.1$ | MNIST, $\epsilon = 0.2$ | MNIST, $\epsilon = 0.3$ | CIFAR, $\epsilon = 2/255$ | CIFAR, $\epsilon = 8/255$ |
|---|---|---|---|---|---|
| Control | 91.58 | 86.45 | 77.99 | 45.53 | 7.09 |
| +RS | 94.33 | 89.79 | 80.68 | 45.93 | 20.27 |

# 4 EXPERIMENTS

## 4.1 EXPERIMENTS ON MNIST AND CIFAR

In addition to the experimental results already presented, we compare our control and "+RS" networks with the best available results presented in the state-of-the-art certifiable defenses of Wong et al. (2018), Dvijotham et al. (2018), and Mirman et al. (2018) in Table 3. We compare their test set accuracy, PGD adversarial accuracy (an evaluation of robustness against a strong 40-step PGD adversarial attack), and provable adversarial accuracy. Additionally, to show that our method can scale to larger architectures, we train and verify a "*+RS (Large)*" network for each dataset and $\epsilon$.

In terms of provable adversarial accuracies, on MNIST, our results are significantly better than those of Wong et al. (2018) for larger perturbations of $\epsilon = 0.3$, and comparable for $\epsilon = 0.1$. On CIFAR-10, our method is slightly less effective, perhaps indicating that more unstable ReLUs are necessary to properly learn a robust CIFAR classifier. We also experienced many more instances of the verifier reaching its allotted 120 second time limit on CIFAR, especially for the less ReLU stable control networks. Full experimental details for each model in Tables 1, 2, and 3, including a breakdown of verification solve results (how many images did the verifier **A.** prove robust **B.** find an adversarial example for **C.** time out on), are available in Appendix E.

Table 3: Comparison of test set, PGD adversarial, and provable adversarial accuracy of networks trained with and without RS Loss. We also provide the best available certifiable adversarial and PGD adversarial accuracy of any single models from Wong et al. (2018), Dvijotham et al. (2018), and Mirman et al. (2018) for comparison, and highlight the best provable accuracy for each $\epsilon$.
\* The provable adversarial accuracy for "+RS (Large)" is only computed for the first 1000 images because the verifier performs more slowly on larger models.
\*\* Dvijotham et al. (2018); Mirman et al. (2018) use a slightly smaller $\epsilon = 0.03 = 7.65/255$.
† Mirman et al. (2018) computes results over 500 images instead of all 10000.
†† Mirman et al. (2018) uses a slightly smaller $\epsilon = 0.007 = 1.785/255$.

| Dataset | Epsilon | Training Method | Test Set Accuracy | PGD Adversarial Accuracy | Provable/Certifiable Adversarial Accuracy |
|---|---|---|---|---|---|
| MNIST | $\epsilon = 0.1$ | Control | 98.94% | 95.12% | 91.58% |
| | | +RS | 98.68% | 95.13% | 94.33% |
| | | +RS (Large)* | 98.95% | 96.58% | 95.60% |
| | | Wong et al. | 98.92% | - | 96.33% |
| | | Dvijotham et al. | 98.80% | 97.13% | 95.56% |
| | | Mirman et al.† | 99.00% | 97.60% | **96.60**% |
| MNIST | $\epsilon = 0.2$ | Control | 98.40% | 93.14% | 86.45% |
| | | +RS | 98.10% | 93.14% | **89.79%** |
| | | +RS (Large)* | 98.21% | 94.19% | 89.10% |
| MNIST | $\epsilon = 0.3$ | Control | 97.75% | 91.64% | 77.99% |
| | | +RS | 97.33% | 92.05% | 80.68% |
| | | +RS (Large)* | 97.54% | 93.25% | 59.60% |
| | | Wong et al. | 85.13% | - | 56.90% |
| | | Mirman et al.† | 96.60% | 93.80% | **82.00**% |
| CIFAR | $\epsilon = 2/255$ | Control | 64.64% | 51.58% | 45.53% |
| | | +RS | 61.12% | 49.92% | 45.93% |
| | | +RS (Large)* | 61.41% | 50.61% | 41.40% |
| | | Wong et al. | 68.28% | - | **53.89%** |
| | | Mirman et al.†,†† | 62.00% | 54.60% | 52.20% |
| CIFAR | $\epsilon = 8/255$ | Control | 50.69% | 31.28% | 7.09% |
| | | +RS | 40.45% | 26.78% | 20.27% |
| | | +RS (Large)* | 42.81% | 28.69% | 19.80% |
| | | Wong et al. | 28.67% | - | 21.78% |
| | | Dvijotham et al.** | 48.64% | 32.72% | 26.67% |
| | | Mirman et al.†, ** | 54.20% | 40.00% | **35.20**% |

## 4.2 Experimental Methods and Details

In our experiments, we use robust adversarial training (Goodfellow et al., 2015) against a strong adversary as done in Madry et al. (2018) to train various DNN classifiers. For each setting of dataset (MNIST or CIFAR) and $\epsilon$, we find a suitable weight on RS Loss via line search (See Table 6 in Appendix D). The same weight is used for each ReLU. During training, we used improved IA for ReLU bound estimation for "+RS" models and use naive IA for "+RS (Large)" models because of memory constraints. For ease of comparison, we trained our networks using the same convolutional DNN architecture as in Wong et al. (2018). This architecture uses two 2x2 strided convolutions with 16 and 32 filters, followed by a 100 hidden unit fully connected layer. For the larger architecture, we also use the same "large" architecture as in Wong et al. (2018). It has 4 convolutional layers with 32, 32, 64, and 64 filters, followed by 2 fully connected layers with 512 hidden units each.

For verification, we used the most up-to-date version of the exact verifier from Tjeng et al. (2017). Model solves were parallelized over 8 CPU cores. When verifying an image, the verifier of Tjeng et al. (2017) first builds a model, and second, solves the verification problem (See Appendix D.2 for details). We focus on reporting solve times because that is directly related to the task of verification itself. All build times for the control and "+RS" models on MNIST that we presented were between 4 and 10 seconds, and full results on build times are also presented in Appendix E.

Additional details on our experimental setup (e.g. hyperparameters) can be found in Appendix D.

## 5 CONCLUSION

In this paper, we use the principle of co-design to develop training methods that emphasize verification as a goal, and we show that they make verifying the trained model much faster. We first demonstrate that natural regularization methods already make the exact verification problem significantly more tractable. Subsequently, we introduce the notion of ReLU stability for networks, present a method that improves a network's ReLU stability, and show that this improvement makes verification an additional 4–13x faster. Our method is universal, as it can be added to any training procedure and should speed up any exact verification procedure, especially MILP-based methods.

Prior to our work, exact verification seemed intractable for all but the smallest models. Thus, our work shows progress toward reliable models that can be proven to be robust, and our techniques can help scale verification to even larger networks.

Many of our methods appear to compress our networks into more compact, simpler forms. We hypothesize that the reason that regularization methods like RS Loss can still achieve very high accuracy is that most models are overparametrized in the first place. There exist clear parallels between our methods and techniques in model compression (Han et al., 2016; Cheng et al., 2017b) – therefore, we believe that drawing upon additional techniques from model compression can further improve the ease-of-verification of networks. We also expect that there exist objectives other than weight sparsity and ReLU stability that are important for verification speed. If so, further exploring the principle of co-design for those objectives is an interesting future direction.

### ACKNOWLEDGEMENTS

This work was supported by the NSF Graduate Research Fellowship under Grant No. 1122374, by the NSF grants CCF-1553428 and CNS-1815221, and by Lockheed Martin Corporation under award number RPP2016-002. We would like to thank Krishnamurthy Dvijotham, Ludwig Schmidt, Michael Sun, Dimitris Tsipras, and Jonathan Uesato for helpful discussions.

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

APPENDIX

## A    NATURAL IMPROVEMENTS

### A.1    NATURAL REGULARIZATION FOR INDUCING WEIGHT SPARSITY

All of the control and "+RS" networks in our paper contain natural improvements that improve weight sparsity, which reduce the number of variables in the LPs solved by the verifier. We observed that the two techniques we used for weight sparsity ($\ell_1$-regularization and small weight pruning) don't hurt test set accuracy but they dramatically improve provable adversarial accuracy and verification speed.

1. $\ell_1$-regularization: We use a weight of $2e-5$ on MNIST and a weight of $1e-5$ on CIFAR. We chose these weights via line search by finding the highest weight that would not hurt test set accuracy.
2. Small weight pruning: Zeroing out weights in a network that are very close to zero. We choose to prune weights less than $1e-3$.

### A.2    A BASIC IMPROVEMENT FOR INDUCING ReLU STABILITY: ReLU PRUNING

We also use a basic idea to improve ReLU stability, which we call ReLU pruning. The main idea is to prune away ReLUs that are not necessary.

We use a heuristic to test whether a ReLU in a network is necessary. Our heuristic is to count how many training inputs cause the ReLU to be active or inactive. If a ReLU is active (the pre-activation satisfies $\hat{z}_{ij}(x) > 0$) for every input image in the training set, then we can replace that ReLU with the identity function and the network would behave in exactly the same way for all of those images. Similarly, if a ReLU is inactive ($\hat{z}_{ij}(x) < 0$) for every training image, that ReLU can be replaced by the zero function.

Extending this idea further, we expect that ReLUs that are *rarely* used can also be removed without significantly changing the behavior of the network. If only a small fraction (say, $10\%$) of the input images activate a ReLU, then replacing the ReLU with the zero function will only slightly change the network's behavior and will not affect the accuracy too much. We provide experimental evidence of this phenomenon on an adversarially trained ($\epsilon = 0.1$) MNIST model. Conservatively, we decided that pruning away ReLUs that are active on less than $10\%$ of the training set or inactive on less than $10\%$ of the training set was reasonable.

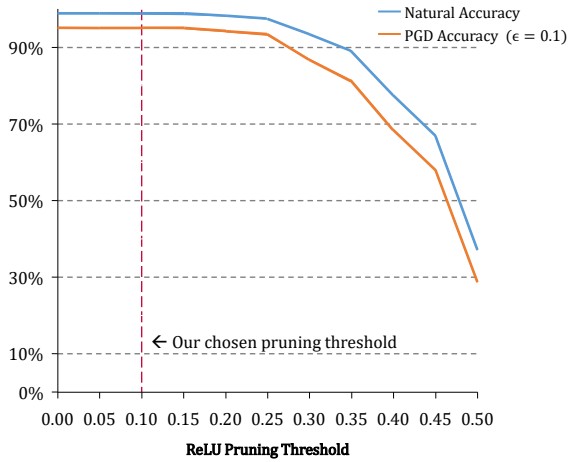

Figure 3: Removing some ReLUs does not hurt test set accuracy or accuracy against a PGD adversary

## B    ADVERSARIAL TRAINING AND WEIGHT SPARSITY

It is worth noticing that adversarial training against $\ell_\infty$ norm-bound adversaries alone already makes networks easier to verify by implicitly improving weight sparsity. Indeed, this can be shown clearly in the case of linear networks. Recall that a linear network can be expressed as $f(x) = Wx + b$. Thus, an $\ell_\infty$ norm-bound perturbation of the input $x$ will produce the output

$$
\begin{aligned}
f(x') &= x'W + b \\
&= xW + b + (x' - x)W \\
&\leq f(x) + \epsilon ||W||_1
\end{aligned}
$$

where the last inequality is just Hölder's inequality. In order to limit the adversary's ability to perturb the output, adversarial training needs to minimize the $||W||_1$ term, which is equivalent to $\ell_1$-regularization and is known to promote weight sparsity (Tibshirani, 1994). Relatedly, Goodfellow et al. (2015) already pointed out that adversarial attacks against linear networks will be stronger when the $\ell_1$-norm of the weight matrices is higher.

Even in the case of nonlinear networks, adversarial training has experimentally been shown to improve weight sparsity. For example, models trained according to Madry et al. (2018) and Wong and Kolter (2018) often learn many weight-sparse layers, and we observed similar trends in the models we trained. However, it is important to note that while adversarial training alone does improve weight sparsity, it is not sufficient by itself for efficient exact verification. Additional regularization like $\ell_1$-regularization and small weight pruning further promotes weight sparsity and gives rise to networks that are much easier to verify.

## C  INTERVAL ARITHMETIC

### C.1  NAIVE INTERVAL ARITHMETIC

Naive IA determines upper and lower bounds for a layer based solely on the upper and lower bounds of the previous layer.

Define $W^+ = \max(W, 0)$, $W^- = \min(W, 0)$, $u = \max(\hat{u}, 0)$, and $l = \max(\hat{l}, 0)$. Then the bounds on the pre-activations of layer $i$ can be computed as follows:

$$\hat{u}_i = u_{i-1} W_i^+ + l_{i-1} W_i^- + b_i \tag{7}$$

$$\hat{l}_i = l_{i-1} W_i^+ + u_{i-1} W_i^- + b_i \tag{8}$$

As noted in Tjeng et al. (2017) and also Dvijotham et al. (2018), this method is efficient but can lead to relatively conservative bounds for deeper networks.

### C.2  IMPROVED INTERVAL ARITHMETIC

We improve upon naive IA by exploiting ReLUs that we can determine to always be active. This allows us to cancel symbols that are equivalent that come from earlier layers of a network.

We will use a basic example of a neural network with one hidden layer to illustrate this idea. Suppose that we have the scalar input $z_0$ with $l_0 = 0, u_0 = 1$, and the network has the following weights and biases:

$$W_1 = \begin{bmatrix} 1 & -1 \end{bmatrix}, \quad b_1 = \begin{bmatrix} 2 & 2 \end{bmatrix}, \quad W_2 = \begin{bmatrix} 1 \\ 1 \end{bmatrix}, \quad b_2 = 0$$

Naive IA for the first layer gives $\hat{l}_1 = l_1 = \begin{bmatrix} 2 & 1 \end{bmatrix}$, $\hat{u}_1 = u_1 = \begin{bmatrix} 3 & 2 \end{bmatrix}$, and applying naive IA to the output $\hat{z}_2$ gives $\hat{l}_2 = 3, \hat{u}_2 = 5$. However, because $\hat{l}_1 > 0$, we know that the two ReLUs in the hidden layer are always active and thus equivalent to the identity function. Then, the output is

$$\hat{z}_2 = z_{11} + z_{12} = \hat{z}_{11} + \hat{z}_{12} = (z_0 + 2) + (-z_0 + 2) = 4$$

Thus, we can obtain the tighter bounds $\hat{l}_2 = \hat{u}_2 = 4$, as we are able to cancel out the $z_0$ terms.

We can write this improved version of IA as follows. First, letting $W_k$ denote row $k$ of matrix $W$, we can define the "active" part of $W$ as the matrix $W_A$, where

$$(W_A)_k = \begin{cases} W_k & \text{if } \hat{l}_{i-1} > 0 \\ 0 & \text{if } \hat{l}_{i-1} \leq 0 \end{cases}$$

Define the "non-active" part of $W$ as

$$W_N = W - W_A$$

Then, using the same definitions for the notation $W^+, W^-, u, l$ as before, we can write down the following improved version of IA which uses information from the previous 2 layers.

$$\hat{u}_i = u_{i-1} W_{iN}^+ + l_{i-1} W_{iN}^- + b_i$$
$$+ u_{i-2} (W_{i-1} W_{iA})^+ + l_{i-2} (W_{i-1} W_{iA})^- + b_{i-1} W_{iA}$$
$$\hat{l}_i = l_{i-1} W_{iN}^+ + u_{i-1} W_{iN}^- + b_i$$
$$+ l_{i-2} (W_{i-1} W_{iA})^+ + u_{i-2} (W_{i-1} W_{iA})^- + b_{i-1} W_{iA}$$

We are forced to to use $l_{i-1,j}$ and $u_{i-1,j}$ if we can not determine whether or not the ReLU corresponding to the activation $z_{i-1,j}$ is active, but we use $l_{i-2}$ and $u_{i-2}$ whenever possible.

We now define some additional notation to help us extend this method to any number of layers. We now seek to define $f_n$, which is a function which takes in four sequences of length $n$ – upper bounds, lower bounds, weights, and biases – and outputs the current layer's upper and lower bounds.

What we have derived so far from (7) and (8) is the following

$$f_1(u_{i-1}, l_{i-1}, W_i, b_i) = (u_{i-1} W_i^+ + l_{i-1} W_i^- + b_i, l_{i-1} W_i^+ + u_{i-1} W_i^- + b_i)$$

Let $\mathbf{u}$ denote a sequence of upper bounds. Let $\mathbf{u}_z$ denote element $z$ of the sequence, and let $\mathbf{u}_{[z:]}$ denote the sequence without the first $z$ elements. Define notation for $\mathbf{l}$, $\mathbf{W}$, and $\mathbf{b}$ similarly.

Then, using the fact that $W_N Z = (WZ)_N$ and $W_A Z = (WZ)_A$, we can show that the following recurrence holds:

$$
\begin{aligned}
f_{n+1}(\mathbf{u}, \mathbf{l}, \mathbf{W}, \mathbf{b}) = \; & f_1(\mathbf{u}_1, \mathbf{l}_1, \mathbf{W}_{1N}, \mathbf{b}_1) \\
& + f_n(\mathbf{u}_{[1:]}, \mathbf{l}_{[1:]}, (\mathbf{W}_2 \mathbf{W}_{1A}, \mathbf{W}_{[2:]}), (\mathbf{b}_2 \mathbf{W}_{1A}, \mathbf{b}_{[2:]}))
\end{aligned}
\tag{9}
$$

Let $\mathbf{u}_{(\mathbf{x},\mathbf{y})}$ denote the sequence $(u_x, u_{x-1}, \cdots, u_y)$, and define $\mathbf{l}_{(\mathbf{x},\mathbf{y})}$, $\mathbf{W}_{(\mathbf{x},\mathbf{y})}$, and $\mathbf{b}_{(\mathbf{x},\mathbf{y})}$ similarly. Then, if we want to compute the bounds on layer $k$ using all information from the previous $k$ layers, we simply have to compute $f_k(\mathbf{u}_{(\mathbf{k-1},\mathbf{0})}, \mathbf{l}_{(\mathbf{k-1},\mathbf{0})}, \mathbf{W}_{(\mathbf{k},\mathbf{1})}, \mathbf{b}_{(\mathbf{k},\mathbf{1})})$.

From the recurrence 9, we see that using information from all previous layers to compute bounds for layer $k$ takes $O(k)$ matrix-matrix multiplications. Thus, using information from all previous layers to compute bounds for all layers of a $d$ layer neural network only involves $O(d^2)$ additional matrix multiplications, which is still reasonable for most DNNs. This method is still relatively efficient because it only involves matrix multiplications – however, needing to perform matrix-matrix multiplications as opposed to just matrix-vector multiplications results in a slowdown and higher memory usage when compared to naive IA. We believe the improvement in the estimate of ReLU upper and lower bounds is worth the time trade-off for most networks.

## C.3 EXPERIMENTAL RESULTS ON IMPROVED IA AND NAIVE IA

In Table 4, we show empirical evidence that the number of unstable ReLUs in each layer of a MNIST network, as estimated by improved IA, tracks the number of unstable ReLUs determined by the exact verifier quite well. We also present estimates determined via naive IA for comparison.

| Dataset | Epsilon | Training Method | Estimation Method | Unstable ReLUs in 1st Layer | Unstable ReLUs in 2nd Layer | Unstable ReLUs in 3rd Layer |
|---|---|---|---|---|---|---|
| MNIST | $\epsilon = 0.1$ | Control | Exact | 61.14 | 185.30 | 31.73 |
| | | | Improved IA | 61.14 | 185.96 (+0.4%) | 43.40 (+36.8%) |
| | | | Naive IA | 61.14 | 188.44 (+1.7%) | 69.96 (+120.5%) |
| | | +RS | Exact | 21.64 | 64.73 | 14.67 |
| | | | Improved IA | 21.64 | 64.80 (+0.1%) | 18.97 (+29.4%) |
| | | | Naive IA | 21.64 | 65.34 (+0.9%) | 33.51 (+128.5%) |
| MNIST | $\epsilon = 0.2$ | Control | Exact | 17.47 | 142.95 | 37.92 |
| | | | Improved IA | 17.47 | 142.95 | 48.88 (+28.9%) |
| | | | Naive IA | 17.47 | 142.95 | 69.75 (+84.0%) |
| | | +RS | Exact | 29.91 | 54.47 | 24.05 |
| | | | Improved IA | 29.91 | 54.47 | 28.40 (+18.1%) |
| | | | Naive IA | 29.91 | 54.47 | 40.47 (+68.3%) |
| MNIST | $\epsilon = 0.3$ | Control | Exact | 36.76 | 83.42 | 40.74 |
| | | | Improved IA | 36.76 | 83.44 (+0.02%) | 46.00 (+12.9%) |
| | | | Naive IA | 36.76 | 83.52 (+0.1%) | 48.27 (+18.5%) |
| | | +RS | Exact | 24.43 | 48.47 | 28.64 |
| | | | Improved IA | 24.43 | 48.47 | 31.19 (+8.9%) |
| | | | Naive IA | 24.43 | 48.47 | 32.13 (+12.2%) |

Table 4: Comparison between the average number of unstable ReLUs as found by the exact verifier of Tjeng et al. (2017) and the estimated average number of unstable ReLUs found by improved IA and naive IA. We compare these estimation methods on the control and "+RS" networks for MNIST that we described in Section 3.3.4

### C.4 ON THE CONSERVATIVE NATURE OF IA BOUNDS

The upper and lower bounds we compute on each ReLU via either naive IA or improved IA are conservative. Thus, every unstable ReLU will always be correctly labeled as unstable, while stable ReLUs can be labeled as either stable or unstable. Importantly, every unstable ReLU, as estimated by IA bounds, is correctly labeled and penalized by RS Loss. The trade-off is that stable ReLUs mislabeled as unstable will also be penalized, which can be an unnecessary regularization of the model.

In Table 5 we show empirically that we can achieve the following two objectives at once when using RS Loss combined with IA bounds.

1. Reduce the number of ReLUs *labeled* as unstable by IA, which is an upper bound on the true number of unstable ReLUs as determined by the exact verifier.

2. Achieve similar test set accuracy and PGD adversarial accuracy as a model trained without RS Loss.

| Dataset | Epsilon | Training Method | Estimation Method | Total Labeled Unstable ReLUs | Test Set Accuracy | | PGD Adversarial Accuracy | |
|---|---|---|---|---|---|---|---|---|
| MNIST | $\epsilon = 0.1$ | Control | Improved IA | 290.5 | 98.94% | | 95.12% | |
| | | +RS | Improved IA | 105.4 | 98.68% | (-0.26%) | 95.13% | (+0.01%) |
| | | Control (Large) | Naive IA | 835.8 | 99.04% | | 96.32% | |
| | | +RS (Large) | Naive IA | 150.3 | 98.95% | (-0.09%) | 96.58% | (+0.26%) |

Table 5: The addition of RS Loss results in far fewer ReLUs labeled as unstable for both 3-layer and 6-layer (Large) networks. The decrease in test set accuracy as a result of this regularization is small.

Even though IA bounds are conservative, these results show that it is still possible to decrease the number of ReLUs labeled as unstable by IA without significantly degrading test set accuracy. When comparing the Control and "+RS" networks for MNIST and $\epsilon = 0.1$, adding RS Loss decreased the average number of ReLUs labeled as unstable (using bounds from Improved IA) from 290.5 to 105.4 with just a 0.26% loss in test set accuracy. The same trend held for deeper, 6-layer networks, even when the estimation method for upper and lower bounds was the more conservative Naive IA.

## D  FULL EXPERIMENTAL SETUP

### D.1  NETWORK TRAINING DETAILS

In our experiments, we use robust adversarial training (Goodfellow et al., 2015) against a strong adversary as done in Madry et al. (2018) to train various DNN classifiers. Following the prior examples of Wong and Kolter (2018) and Dvijotham et al. (2018), we introduced a small tweak where we increased the adversary strength linearly from $0.01$ to $\epsilon$ over first half of training and kept it at $\epsilon$ for the second half. We used this training schedule to improve convergence of the training process.

For MNIST, we trained for $70$ epochs using the Adam optimizer (Kingma and Ba, 2015) with a learning rate of $1e{-}4$ and a batch size of $32$. For CIFAR, we trained for $250$ epochs using the Adam optimizer with a learning rate of $1e{-}4$. When using naive IA, we used a batch size of $128$, and when using improved IA, we used a batch size of $16$. We used a smaller batch size in the latter case because improved IA incurs high RAM usage during training. To speed up training on CIFAR, we only added in RS Loss regularization in the last $20\%$ of the training process. Using this same sped-up training method on MNIST did not significantly affect the results.

| Dataset | Epsilon | $\ell_1$ weight | RS Loss weight |
|---------|---------|-----------------|----------------|
| MNIST   | 0.1     | 2e−5            | 12e−5          |
| MNIST   | 0.2     | 2e−5            | 1e−4           |
| MNIST   | 0.3     | 2e−5            | 12e−5          |
| CIFAR   | 2/255   | 1e−5            | 1e−3           |
| CIFAR   | 8/255   | 1e−5            | 2e−3           |

Table 6: Weights chosen using line search for $\ell_1$ regularization and RS Loss in each setting

For each setting, we find a suitable weight on RS Loss via line search. The same weight is used for each ReLU. The five weights we chose are displayed above in Table 6, along with weights chosen for $\ell_1$-regularization.

We also train "+RS" models using naive IA to show that our technique for inducing ReLU stability can work while having small training time overhead – full details on "*+RS (Naive IA)*" networks are in Appendix E.

### D.2  VERIFIER OVERVIEW

The MILP-based exact verifier of Tjeng et al. (2017), which we use, proceeds in two steps for every input. They are the model-build step and the solve step.

First, the verifier builds a MILP model based on the neural network and the input. In particular, the verifier will compute upper and lower bounds on each ReLU using a specific bound computation algorithm. We chose the default bound computation algorithm in the code, which uses LP to compute bounds. LP bounds are tighter than the bounds computed via IA, which is another option available in the verifier. The model-build step's speed appeared to depend primarily on the tightening algorithm (IA was faster than LP) and the number of variables in the MILP (which, in turn, depends on the sparsity of the weights of the neural network). The verifier takes advantage of these bounds by not introducing a binary variables into the MILP formulation if it can determine that a particular ReLU is stable. Thus, using LP as the tightening algorithm resulted in higher build times, but led to easier MILP formulations.

Next, the verifier solves the MILP using an off-the-shelf MILP solver. The solver we chose was the commercial Gurobi Solver, which uses a branch-and-bound method for solving MILPs. The solver's speed appeared to depend primarily on the number of binary variables in the MILP (which corresponds to the number of unstable ReLUs) as well as the total number of variables in the MILP (which is related to the sparsity of the weight matrices). While these two numbers are strongly correlated with solve times, some solves would still take a long time despite having few binary

variables. Thus, understanding what other properties of neural networks correspond to MILPs that are easy or hard to solve is an important area to explore further.

### D.3    VERIFIER DETAILS

We used the most up-to-date version of the exact verifier from Tjeng et al. (2017) using the default settings of the code. We allotted 120 seconds for verification of each input datapoint using the default model build settings. We ran our experiments using the commercial Gurobi Solver (version 7.5.2), and model solves were parallelized over 8 CPU cores with Intel Xeon CPUs @ 2.20GHz processors. We used computers with 8–32GB of RAM, depending on the size of the model being verified. All computers used are part of an OpenStack network.

# E  FULL EXPERIMENTAL VERIFICATION RESULTS

| Dataset | Epsilon | Training Method | Test Set Accuracy | PGD Adversarial Accuracy | Verifier Upper Bound | Provable Adversarial Accuracy | Total Unstable ReLUs | Avg Solve Time (s) | Avg Build Time (s) |
|---|---|---|---|---|---|---|---|---|---|
| MNIST | $\epsilon = 0.1$ | Adversarial Training* | 99.17% | 95.04% | 96.00% | 19.00% | 1517.9 | 2970.43 | 650.93 |
| | | +$\ell_1$-regularization | 99.00% | 95.25% | 95.98% | 82.17% | 505.3 | 21.99 | 79.13 |
| | | +Small Weight Pruning | 98.99% | 95.38% | 94.93% | 89.13% | 502.7 | 11.71 | 19.30 |
| | | +ReLU Pruning (Control) | 98.94% | 95.12% | 94.45% | 91.58% | 278.2 | 6.43 | 9.61 |
| | | +RS | 98.68% | 95.13% | 94.38% | 94.33% | 101.0 | 0.49 | 4.98 |
| | | +RS (Naive IA) | 98.53% | 94.86% | 94.54% | 94.32% | 158.3 | 0.96 | 4.82 |
| | | +RS (Large)** | 98.95% | 96.58% | 95.60% | 95.60% | 119.5 | 0.27 | 156.74 |
| MNIST | $\epsilon = 0.2$ | Control | 98.40% | 93.14% | 90.71% | 86.45% | 198.3 | 9.41 | 7.15 |
| | | +RS | 98.10% | 93.14% | 89.98% | 89.79% | 108.4 | 1.13 | 4.43 |
| | | +RS (Naive IA) | 98.08% | 91.68% | 88.87% | 85.54% | 217.2 | 8.50 | 4.67 |
| | | +RS (Large)** | 98.21% | 94.19% | 90.40% | 89.10% | 133.0 | 2.93 | 171.10 |
| | | Wong et al. (2018)*** | 95.06% | 89.03% | - | 80.29% | - | - | - |
| MNIST | $\epsilon = 0.3$ | Control | 97.75% | 91.64% | 83.83% | 77.99% | 160.9 | 11.80 | 5.14 |
| | | +RS | 97.33% | 92.05% | 81.70% | 80.68% | 101.5 | 2.78 | 4.34 |
| | | +RS (Naive IA) | 97.06% | 89.19% | 79.12% | 76.70% | 179.0 | 6.43 | 4.00 |
| | | +RS (Large)** | 97.54% | 93.25% | 83.70% | 59.60% | 251.2 | 37.45 | 166.39 |
| CIFAR | $\epsilon = 2/255$ | Control | 64.64% | 51.58% | 50.23% | 45.53% | 360.0 | 21.75 | 66.42 |
| | | +RS | 61.12% | 49.92% | 47.79% | 45.93% | 234.3 | 13.50 | 52.58 |
| | | +RS (Naive IA) | 57.83% | 47.03% | 45.33% | 44.44% | 170.1 | 6.30 | 47.11 |
| | | +RS (Large)** | 61.41% | 50.61% | 51.00% | 41.40% | 196.7 | 29.88 | 335.97 |
| CIFAR | $\epsilon = 8/255$ | Control | 50.69% | 31.28% | 33.46% | 7.09% | 665.9 | 82.91 | 73.28 |
| | | +RS | 40.45% | 26.78% | 22.74% | 20.27% | 54.2 | 22.33 | 38.84 |
| | | +RS (Naive IA) | 46.19% | 29.66% | 26.07% | 18.90% | 277.8 | 33.63 | 23.66 |
| | | +RS (Large)** | 42.81% | 28.69% | 25.20% | 19.80% | 246.5 | 20.14 | 401.72 |

Table 7: Full results on natural improvements from Table 1, control networks (which use all of the natural improvements and ReLU pruning), and "+RS" networks from Tables 2 and 3. While we are unable to determine the true adversarial accuracy, we provide two upper bounds and a lower bound. Evaluations of robustness against a strong 40-step PGD adversary (PGD adversarial accuracy) gives one upper bound, and the verifier itself gives another upper bound because it can also prove that the network is *not robust* to perturbations on certain inputs by finding adversarial examples. The verifier simultaneously finds the provable adversarial accuracy, which is a lower bound on the true adversarial accuracy. The gap between the verifier upper bound and the provable adversarial accuracy (verifier lower bound) corresponds to inputs that the verifier times out on. These are inputs that the verifier can not prove to be robust *or* not robust in 120 seconds. Build times and solve times are reported in seconds. Finally, average solve time includes timeouts. In other words, verification solves that time out contribute 120 seconds to the total solve time.

* The "Adversarial Training" network uses a 3600 instead of 120 second timeout and is only verified for the first 100 images because verifying it took too long.

** The "+RS (Large)" networks are only verified for the first 1000 images because of long build times.

*** Wong et al. (2018); Dvijotham et al. (2018), and Mirman et al. (2018), which we compare to in Table 3, do not report results on MNIST, $\epsilon = 0.2$ in their papers. We ran the publicly available code of Wong et al. (2018) on MNIST, $\epsilon = 0.2$ to generate these results for comparison.

## F    DISCUSSION ON VERIFICATION AND CERTIFICATION

Exact verification and certification are two related approaches to formally verifying properties of neural networks, such as adversarial robustness. In both cases, the end goal is formal verification. Certification methods, which solve an easier-to-solve relaxation of the exact verification problem, are important developments because exact verification previously appeared computationally intractable for all but the smallest models.

For the case of adversarial robustness, certification methods exploit a trade-off between provable robustness and speed. They can fail to provide certificates of robustness for some inputs that are actually robust, but they will either find or fail to find certificates of robustness quickly. On the other hand, exact verifiers will always give the correct answer if given enough time, but exact verifiers can sometimes take many hours to formally verify robustness on even a single input.

In general, the process of training a robust neural network and then formally verifying its robustness happens in two steps.

- Step 1: Training
- Step 2: Verification or Certification

Most papers on certification, including Wong and Kolter (2018); Wong et al. (2018); Dvijotham et al. (2018); Raghunathan et al. (2018) and Mirman et al. (2018), propose a method for step 2 (the certification step), and then propose a training objective in step 1 that is directly related to their method for step 2. We call this paradigm "co-training." In Raghunathan et al. (2018), they found that using their step 2 on a model trained using Wong and Kolter (2018)'s step 1 resulted in extremely poor provable robustness (less than 10%), and the same was true when using Wong and Kolter (2018)'s step 2 on a model trained using their step 1.

We focus on MILP-based exact verification as our step 2, which encompasses the best current exact verification methods. The advantage of using exact verification for step 2 is that it will be accurate, regardless of what method is used in step 1. The disadvantage of using exact verification for step 2 is that it could be extremely slow. For our step 1, we used standard robust adversarial training. In order to significantly speed up exact verification as step 2, we proposed techniques that could be added to step 1 to induce weight sparsity and ReLU stability.

In general, we believe it is important to develop effective methods for step 1, given that step 2 is exact verification. However, ReLU stability can also be beneficial for tightening the relaxation of certification approaches like that of Wong et al. (2018) and Dvijotham et al. (2018), as unstable ReLUs are the primary cause of the overapproximation that occurs in the relaxation step. Thus, our techniques for inducing ReLU stability can be useful for certification as well.

Finally, in recent literature on verification and certification, most works have focused on formally verifying the property of adversarial robustness of neural networks. However, verification of other properties could be useful, and our techniques to induce weight sparsity and ReLU stability would still be useful for verification of other properties for the exact same reasons that they are useful in the case of adversarial robustness.

