# OpenReview forum: "Training for Faster Adversarial Robustness Verification via Inducing ReLU Stability"
_ICLR.cc/2019/Conference_

### Official Review · AnonReviewer2 · 2018-10-16
**Good paper, but the value of "verification" over "certification" is not clear**

**Rating:** 5
**Confidence:** 3

**Review:**

The paper presents several ways to regularize plain ReLU networks to optimize 3 things

- the adversarial robustness, defined as the fraction of examples for which adversarial perturbation exists
- the provable adversarial robustness, defined as the fraction of examples for which some method can show that there exists no adversarial example within a certain time budget
- the verification speed, i.e. the amount of time it takes some method to verify whether there is an adversarial example or not

Overall, the ideas are sound and the analysis is solid. My main concern is the comparison between the authors method and the 'certification' methods, both conceptually and regarding performance.

The authors note that their method falls under 'verification', whereas many competing methods fall under 'certification'. They point to two advantages of verification over certification: (1) the ability to provide true negatives, i.e. prove that an adversarial example exists when it does, and (2) certification requires that 'models must be trained and optimized for a specific certification method'. However, neither argument convinces me regarding the utility of the authors method.

Regarding (2): The authors method also requires training the network in a specific way (with RS loss), and it is only compatible with verifiers that care about ReLU stability.

Regarding (1): It is not clear that this would be helpful at all. Is it really that much better if method A has 80% proven robustness and 20% proven non-robustness versus method B that has 80% proven robustness and 20% unknown? One could make the case that method B is actually even better.

So overall, I think one has to compare the authors method and the certification methods head-to-head. And in table 3, where this is done, Dvijotham comes out on top 2 out of 2 times and Wong comes out on top 2 out of 4 times. That does not seem convincing. Also, what about the performance numbers form other papers discussed in section 2?

-------

Other issues:

At first glance, the fact that the paper only deals with (small) plain ReLU networks seems to be a huge downside. While I'm not familiar with the verification / certification literature, from reading the paper, I suspect that all the other verification / certification methods also only deal with that or highly similar architectures. However, I will defer to the other reviewers if this is not the case.

To expand upon my comment above, I think the paper should discuss true adversarial accuracy on top of provable adversarial robustness. Looking at table 1, for instance, for rows 2, 3 and 4, it seems that the verifier used much less than 120 seconds on average. Does that mean the verifier finished for all test examples? And wouldn't that mean that the verifier determined for each test example exactly whether an adversarial example existed or not? In that case, I would write "true adversarial accuracy" instead of "provable adversarial accuracy" as column header. If the verifiers did not finish, I would include in the paper for how many examples the result was "adverarial example exists" and for how many the result was "timeout". I would also include that information in table 3, and I would also include proving / certification times there.

Based on the paper, I'm not quite sure whether the idea of training with L1 regularization and/or small weight pruning and/or ReLU pruning for the purpose of improving robustness / verifiability was an original idea of this paper. In either case, this should be made clear. Also, the paper seems to use networks with adversarial training, small weight pruning, L1 and ReLU pruning as its baseline in most cases (all figures except table 1). If some of these techniques are original contributions, this might not be an appropriate baseline to use, even if it is a strong baselines.

Why are most experiments presented outside of the "experiments" section? This seems to be bad presentation.

I would include all test set accuracy values instead of writing "its almost as high". Also, in table 3, it appears as if using RS loss DOES in fact reduce test error significantly, at least for CIFAR. Why is that?

While, again, I'm not familiar with the background work on verification / certification, it appears to me from reading this paper that all known verification algorithms perform terribly and are restricted to a narrow range of network architectures. If that is the case, one has to wonder whether that line of research should be encouraged to continue.

--------

Minor issues:

- "our focus will be on the most common architecture for state-of-the-art models: k-layer fully-connected feed-forward DNN classifiers" Citation needed. Otherwise, I would suggest removing this statement.
- "such models can be viewed as a function f(.,W)" - you also need to include the bias in the formula I think
- "convolutional layers can be represented as fully-connected layers". I think what you mean is "convolutional layers can be represented as matrix multiplication"
- could you make the difference between co-design and co-training more clear?
- The paper could include in the appendix a section outlining the verification method of Tjeng

---

> ### Author Response · Authors · 2018-11-13
> **Response (1/3)**
>
> We thank the reviewer for their detailed comments; they will be helpful in revising our manuscript.
>
> We appreciate the point about the importance of “verification” compared to “certification” as it is indeed a great question!
> Now, one should note that, in this context, formal verification is the ultimate end goal we strive for; certification is just a fast “shortcut” that can get us closer to this goal, at the expense of sacrificing part of the robustness guarantee.
> Of course, if certification already gives us a satisfactory level of robustness, this tradeoff can be beneficial. However, it is unclear if the current state-of-the-art (SOTA) certification methods like Wong et. al 2018 [1] are at this point able to deliver such robustness once we move beyond the smallest perturbation sizes (see the MNIST, eps=0.3 case in Table 3 of our manuscript). Additionally, when applied to neural networks not specifically trained for that certification method, they give vacuous bounds [2].
> Thus, as of now, using and improving formal verification methods should still be our focus.
>
> -------------------------------------------------------------------------------------------------------------------------------------
>
> Now, we would like to address your high-level comments in more detail, followed by your specific comments:
>
> > Regarding (2): The authors method also requires training the network in a specific way (with RS loss), and it is only compatible with verifiers that care about ReLU stability.
>
> We view RS Loss as a regularization method, similar to L1 regularization. It can be added to any training procedure, and it is designed for a natural goal - encouraging stable ReLUs.
>
> Even if one could, in principle, imagine a verification approach that does not benefit from the natural goal of ReLU stability, all effective verification methods that we are aware of, either falling under the broader class of SMT-based verifiers [3,4] or MIP-based verifiers [5,6,7], can benefit from ReLU stability. For example, [3] states that “When tighter bounds are derived for ReLU variables, these variables can sometimes be eliminated, i.e., fixed to the active or inactive state, without splitting.” [6] writes: “we conjecture that the large increase of binary variables in the problem caused by the binary constraints on the input creates the large performance gap between the Reuters dataset and MNIST.” (the Reuters dataset had more binary variables and took much longer to verify)
>
> > Regarding (1): While the authors method can provide true negatives, they are not discussed in the paper at all.
>
> > I would include in the paper for how many examples the result was "adverarial example exists" and for how many the result was "timeout".
>
> We do discuss upper bounds for the true robustness of all of our models, as well as the number of timeouts, in Appendix E. We appreciate that you bring up this point though, and we will work to point readers to these relevant details in Appendix E from the main body of the paper when revising.
>
> In Appendix E, the column labeled “Verifier Upper Bound” is simply 100% minus the number of true negatives (“adversarial examples exists” cases) - it describes the maximum possible value for the true adversarial accuracy. The difference between the upper bound (“Verifier Upper Bound”) and the lower bound (“Provable Adversarial Accuracy”) on the true adversarial accuracy equals how many examples reached their “timeout,” as we can not determine which category they belong to.
>
> > At first glance, the fact that the paper only deals with (small) plain ReLU networks seems to be a huge downside.
>
> > While, again, I'm not familiar with the background work on verification / certification, it appears to me from reading this paper that all known verification algorithms perform terribly and are restricted to a narrow range of network architectures.
>
> Indeed, we purposely chose our architectures to match those of prior works in certification literature for the fairest possible comparison.
>
> We agree that expanding beyond our current capabilities for verification and certification is an important direction for further research. Our contribution in this manuscript is to show that training for ease-of-verification via inducing weight sparsity and ReLU stability can help scale verification. Prior to our work, verification methods struggled for neural networks with just a few hundreds ReLUs in total. Using our methods, networks as large as our “large” convolutional CIFAR network, which have over 60000 ReLUs (most of which can be made stable), can be verified.

---

> > ### Author Response · Authors · 2018-11-13
> > **Response (2/3)**
> >
> > Finally, we would like to address your more specific comments:
> >
> > > Is it really that much better if method A has 80% proven robustness and 20% proven non-robustness versus method B that has 80% proven robustness and 20% unknown? One could make the case that method B is actually even better.
> >
> > With regards to the example you describe involving method A and method B, we would argue that the results of method A are better. It is better to know exactly in which cases your model is robust and in which cases it is not. If model B is close to 20% non-robust (if all or most of the 20% unknown cases are non-robust), it is better to know than not to know.
> >
> > If you can get better proven robustness with method B and method A is fundamentally limited, then perhaps method B is better. But in the case where method A is verification and method B is certification, as of now, method B does not seem to offer significant advantages in robustness, and method A seems to have limitations that can be addressed - our work is one step toward doing so.
> >
> > > Also, what about the performance numbers form other papers discussed in section 2?
> >
> > The performance of other papers discussed in Section 2 (related works) are worse than the results of Wong [1] and Dvijotham [8], which is why we primarily compare to those two works. In comparisons to [1] and [8], our results are within a few % or better.
> >
> > > Looking at table 1, for instance, for rows 2, 3 and 4, it seems that the verifier used much less than 120 seconds on average. Does that mean the verifier finished for all test examples?
> >
> > Rows 2, 3, and 4 in Table 1 show the average verification time over all test examples, given that the “timeout” is 120 seconds. We count examples that “timeout” as taking 120 seconds. Thus, the average will always be less than 120 seconds, and we observe that this average decreases as we use more natural regularization techniques during training. All of Rows 2, 3, and 4 have some proportion of test examples where the verifier reaches timeout and does not finish (13.81%, 5.8%, 2.87%, respectively - cf. Appendix E).
> >
> > > Based on the paper, I'm not quite sure whether the idea of training with L1 regularization and/or small weight pruning and/or ReLU pruning for the purpose of improving robustness / verifiability was an original idea of this paper. In either case, this should be made clear. Also, the paper seems to use networks with adversarial training, small weight pruning, L1 and ReLU pruning as its baseline in most cases (all figures except table 1). If some of these techniques are original contributions, this might not be an appropriate baseline to use, even if it is a strong baselines.
> >
> > L1 regularization, weight pruning, and ReLU pruning are original contributions in terms of their application to improving the ease-of-verification of networks. We felt that these ideas are natural and/or commonly used in other settings, and thus did not want to over-emphasize our applying them as major contributions.
> >
> > We chose to use a strong baseline as our control in order to isolate the effect of adding RS Loss. Simply using adversarial training as a baseline vs. our final +RS network (which includes adversarial training, L1, small weight pruning, ReLU pruning, and RS Loss) would show a drastic speedup gain in support of the combination of all of our methods, but it would not isolate the effect of each individual method.
> >
> > > Why are most experiments presented outside of the "experiments" section? This seems to be bad presentation.
> >
> > We aim to present the most relevant experimental results that support our claims in the body of the paper, and leave the more in-depth details for the “Experiments” section and the Appendix.
> >
> > > I would include all test set accuracy values instead of writing "its almost as high".
> >
> > This is a good suggestion - we will include the test set accuracy in Fig. 2.
> >
> > > Also, in table 3, it appears as if using RS loss DOES in fact reduce test error significantly, at least for CIFAR. Why is that?
> >
> > For CIFAR, we chose a much higher weight on RS Loss to improve ease-of-verifiability further. We chose weights that did cause test set accuracy degradation to achieve this goal. We remarked in section 4.1 that a potential limitation of our current method is that CIFAR may require more unstable ReLUs to fit properly, which will cause a more noticeable tradeoff between test set accuracy and ReLU stability.

---

> > > ### Author Response · Authors · 2018-11-13
> > > **Response (3/3)**
> > >
> > > > Minor issues:
> > >
> > > > - "our focus will be on the most common architecture for state-of-the-art models: k-layer fully-connected feed-forward DNN classifiers" Citation needed. Otherwise, I would suggest removing this statement.
> > > > - "such models can be viewed as a function f(.,W)" - you also need to include the bias in the formula I think
> > > > - "convolutional layers can be represented as fully-connected layers". I think what you mean is "convolutional layers can be represented as matrix multiplication"
> > > > - could you make the difference between co-design and co-training more clear?
> > > > - The paper could include in the appendix a section outlining the verification method of Tjeng
> > >
> > > These are all helpful comments that we will address while revising. To clarify here, co-design means optimizing for multiple design objectives during training and co-training mean using a specific training procedure combined with a specific certification procedure.
> > >
> > > REFERENCES
> > > -------------------------------------------------------------------------------------------------------------------------------------
> > > All references here are also cited in the submitted paper.
> > >
> > > [1] Eric Wong, Frank Schmidt, Jan Hendrik Metzen, and J. Zico Kolter. Scaling provable adversarial defenses. NIPS, 2018.
> > >
> > > [2] A. Raghunathan, J. Steinhardt, and P. Liang. Certified defenses against adversarial examples. In International Conference on Learning Representations (ICLR), 2018.
> > >
> > > [3] Guy Katz, Clark Barrett, David L. Dill, Kyle Julian, and Mykel J. Kochenderfer. Reluplex: An efficient smt solver for verifying deep neural networks. In Rupak Majumdar and Viktor Kuncak, ˇ editors, Computer Aided Verification, pages 97–117, Cham, 2017. Springer International Publishing. ISBN 978-3-319-63387-9.
> > >
> > > [4] Rudiger Ehlers. Formal verification of piece-wise linear feed-forward neural networks. In Deepak ¨ D’Souza and K. Narayan Kumar, editors, Automated Technology for Verification and Analysis, pages 269–286, Cham, 2017. Springer International Publishing. ISBN 978-3-319-68167-2.
> > >
> > > [5] Vincent Tjeng, Kai Xiao, and Russ Tedrake. Verifying neural networks with mixed integer programming. CoRR, abs/1711.07356, 2017. URL http://arxiv.org/abs/1711.07356.
> > >
> > > [6] Alessio Lomuscio and Lalit Maganti. An approach to reachability analysis for feed-forward relu neural networks. CoRR, abs/1706.07351, 2017. URL http://arxiv.org/abs/1706. 07351.
> > >
> > > [7] Chih-Hong Cheng, Georg Nuhrenberg, and Harald Ruess. Maximum resilience of artificial neural ¨ networks. CoRR, abs/1705.01040, 2017a. URL http://arxiv.org/abs/1705.01040.
> > >
> > > [8] Krishnamurthy Dvijotham, Sven Gowal, Robert Stanforth, Relja Arandjelovic, Brendan O’Donoghue, Jonathan Uesato, and Pushmeet Kohli. Training verified learners with learned verifiers. arXiv preprint arXiv:1805.10265, 2018.

---

> > > > ### Public Comment · (anonymous) · 2018-11-13
> > > > **Certification is not limited!**
> > > >
> > > > Please check out this paper https://arxiv.org/abs/1810.12715
> > > >
> > > > It is quite recent, and I would not expect a comparison or use it was a baseline. But if your argument is certification based methods are limited/not scalable, this paper clearly answers both those questions in the negative. If the claim of your paper is, "Certification is fundamentally limited and hence we need alternative approaches",  I think that claim is not justified in the least, as clearly proven otherwise by the above paper.
> > > >
> > > > Also, these numbers significantly improve over [8] you cite earlier. Please consider toning down your criticism/claims in the light of evidence provided by https://arxiv.org/abs/1810.12715.

---

> > > > > ### Author Response · Authors · 2018-11-13
> > > > > **Response**
> > > > >
> > > > > As far as we can tell, this paper achieves these SOTA results using an exact verification approach as opposed to using certification. Thus, it actually shows that verification can be scaled further, surpassing the results of current SOTA certification. In particular, this supports our idea that improving verification should be our primary focus. Please do correct us if we are misunderstanding the methodology of this paper.
> > > > >
> > > > > We did not intend for our response or paper to focus on the message of "Certification is limited." Instead, our primary message is "Verification has been limited in the past, but it can be improved using our techniques. Our improvements are on par with SOTA certification methods, which are a ‘shortcut’ to fully doing verification." This newer paper https://arxiv.org/pdf/1810.12715.pdf actually improves verification beyond our current results, and we believe that their interval bound propagation method could be used in conjunction with our methods for ReLU stability for even more easily verifiable networks.
> > > > >
> > > > > Lastly, we want to clarify the words "certification" and "verification" one more time just to avoid any confusion about terms. We use "certification" to refer to techniques that use relaxation-based approaches to provide certificates of robustness. These techniques differ from "verification" techniques because of the relaxation step, which is helpful for speeding up the certification process but can result in a decrease in the number of inputs that can be certified.

---

> > > > > > ### Public Comment · (anonymous) · 2018-11-14
> > > > > > **Not quite true**
> > > > > >
> > > > > > The paper achieves these SOTA using a "relaxation-based" method for training, not exact verification. It relaxes every non-linear node in the network using interval analysis, and then uses these bounds to guide the training procedure.
> > > > > >
> > > > > > They use a MILP solver to measure the robustness however. Perhaps a comparison of the run-times for networks trained with "relaxations" vs your approach is useful here?
> > > > > >
> > > > > > I strongly suspect sparse weights/ReLU-stability might already be a consequence of training with relaxations as both of these things tend to make relaxations tighter. The relaxations would be tightest when the network is simply linear in the neighborhood of a point. More local linearity->tighter relaxations and tighter relaxations->easier to optimize.  I am concerned that this is just another path to arriving at weights that have similar properties as the ones discussed in https://arxiv.org/pdf/1711.00851.pdf .
> > > > > >
> > > > > > I do believe exact verification has merit in verifying more generalized properties, but training for "adversarial" robustness seems doable with relaxation based approaches, particularly as the relaxations keep getting tighter!

---

> > > > > > > ### Public Comment · (anonymous) · 2018-11-14
> > > > > > > **Relaxation based training may already induce weight sparsity/ReLU stability**
> > > > > > >
> > > > > > > Check out Figure 8 in https://arxiv.org/pdf/1711.00851.pdf

---

> > > > > > > > ### Author Response · Authors · 2018-11-15
> > > > > > > > **Response**
> > > > > > > >
> > > > > > > > We agree here as well - ReLU stability and weight sparsity are not properties unique to the networks we train. In fact, we specifically acknowledge what you point out here in our Appendix B - that training as in [10], as well as the adversarial training approach of Madry et. al [11], already seem to improve weight sparsity. However, as shown in Table 1, we found that adversarial training alone was not enough for easily verifiable networks, so we used additional regularization for the natural goals of ReLU stability and weight sparsity. All of these methods appear complementary, rather than conflicting.
> > > > > > > >
> > > > > > > > [11] Aleksander Madry, Aleksandar Makelov, Ludwig Schmidt, Dimitris Tsipras, and Adrian Vladu. Towards deep learning models resistant to adversarial attacks. In International Conference on Learning Representations (ICLR), 2018.

---

> > > > > > > > > ### Comment · AnonReviewer2 · 2018-11-19
> > > > > > > > > **Response from reviewer**
> > > > > > > > >
> > > > > > > > > I'm not convinced by your statement "using and improving formal verification methods should still be our focus", and I'm not convinced that knowing that there is an adversarial example has great value. After all, we can simply assume that all examples that are not certified by some certification method have an adversarial example, as a worst case.
> > > > > > > > >
> > > > > > > > > "We view RS Loss as a regularization method, similar to L1 regularization. It can be added to any training procedure, and it is designed for a natural goal - encouraging stable ReLUs." In all experients, RS loss reduced test error. However, the goal of "natural" regularization is to increase test error by reducing the generalization gap. So I don't see how RS loss would compete with L1 regularization or weight decay.
> > > > > > > > >
> > > > > > > > > I'm still unimpressed by your experimental results. If you outperform many of the methods cited in section 2, I would include those results in the paper. Right now, for MNIST with \epsilon=0.2, you don't show any comparable method. Also, if Dijotham et al is SOTA, why not run it yourself for all the scenarios you study? I'm also still unconvinced that verification via linear program solvers is a fruitful direction for research in general as all results presented (both from your method, the baseline and competing papers) seem horrendously bad to me (50% test accuracy on CIFAR ... ?).
> > > > > > > > >
> > > > > > > > > Because L1 regularization, weight pruning, and ReLU pruning are all original contributions; because of Appendix E; because your method is "universal" with regards to current verification methods; because the nets you use are the largest in terms of size within the verification literature; and because of your responsiveness to my and other criticisms overall, I increase my score to 5. I'm not an expert on the topic of the paper and wouldn't mind seeing this paper accepted, or deferring to more knowledgeable reviewers.

---

> > > > > > > > > > ### Author Response · Authors · 2018-11-25
> > > > > > > > > > **Response**
> > > > > > > > > >
> > > > > > > > > > Thank you for taking the time to carefully read through our responses and paper. We appreciate the time and expertise you put into your review.
> > > > > > > > > >
> > > > > > > > > > We still believe that improving verification is an important research topic. We state that "using and improving formal verification methods should still be our focus" because the ultimate end goal is formal verification of properties such as adversarial robustness. If verification was computationally infeasible in most settings, which seemed to be the case prior to our work, then using the “shortcut” of certification would indeed be the only viable approach. However, as we show in our paper,  verification can be made more feasible using our techniques for training easily verifiable models, and thus verification is viable after all. We thus believe improving verification further will lead to better results regarding properties such as adversarial robustness, and, as such, it is an important research topic. Finally, we do not think we have to view verification vs. certification as an either-or “choice.” After all, our technique could potentially improve certification too. As the anonymous commenter pointed out, the certification relaxation of Wong [1, 10] becomes tighter if there are fewer unstable ReLUs.
> > > > > > > > > >
> > > > > > > > > > We chose to compare our technique to SOTA certification results to show its relative effectiveness. We do not believe it makes as much sense to compare with methods that are not SOTA. We do agree though that it is worth explaining that we specifically compare to Wong [1] and Dvijotham [8] as opposed to the other works listed in section 2 specifically because [1] and [8] are SOTA.
> > > > > > > > > >
> > > > > > > > > > You correctly note that other works have not considered the MNIST, eps=0.2 case. Dvijotham [8] does not have publicly available code, while Wong [1] does. When using Wong’s code on the eps=0.2 case a few months ago, we got 80.29% certifiable accuracy, which is lower than our 89.79% provable accuracy. However, we believe it is more fair to present the best results that each original author had previously reported in literature, as we did not want to run their code with incorrect settings. With that said, our best attempts to use Wong’s [1] code can definitely be documented in the Appendix to provide a datapoint for comparison.
> > > > > > > > > >
> > > > > > > > > > We agree that 50% test accuracy on CIFAR is not ideal. In this paper, our focus was on obtaining higher provable adversarial accuracy via verification on CIFAR, as this had not been achieved before. We believe that there is much additional research to be done toward understanding how to obtain more easily verifiable networks without sacrificing as much test accuracy, or toward understanding if/when the tradeoff is necessary. We believe that our work is a step in that direction.

---

> > > > > > > ### Author Response · Authors · 2018-11-15
> > > > > > > **Response (Clarifications on terms)**
> > > > > > >
> > > > > > > We absolutely agree with you on these points, and believe that we had a misunderstanding regarding terminology earlier. Thank you for clarifying; we will clarify here our viewpoint here, which we believe does not conflict with yours.
> > > > > > >
> > > > > > > This is how we view the overall procedure to train a neural network with high true adversarial accuracy, and then verify/certify that accuracy.
> > > > > > >
> > > > > > > Step 1: Training.
> > > > > > > Step 2: Verification/Certification.
> > > > > > >
> > > > > > > It is very difficult to compute the true adversarial accuracy during training. Thus, current training procedures optimize for an approximation of it. In our work, we use standard adversarial training for Step 1, which optimizes for PGD-adversarial accuracy. PGD-adversarial accuracy is an upper-bound approximation of true adversarial accuracy, while “relaxation”-based approaches are a lower-bound approximation of true adversarial accuracy. Our contribution is to add regularization to Step 1 which improves ease-of-verification in Step 2.
> > > > > > >
> > > > > > > To compare the main approaches of the 4 papers being discussed (ours, Gowal et al. [9], and Wong et. al [1] / Dvijotham et. al [8]):
> > > > > > > Our paper does
> > > > > > > Step 1: upper-bound approximation and regularization, Step 2: (MILP) Verification
> > > > > > > [9] does
> > > > > > > Step 1: lower-bound approximation, Step 2: (MILP) Verification
> > > > > > > [1] and [8] do
> > > > > > > Step 1: lower-bound approximation, Step 2: Certification
> > > > > > >
> > > > > > > Our contention regarding verification/certification is simply that we should try to use verification in Step 2. Our work presents one possible Step 1 to make the use of verification in Step 2 easier. The work of [9], which was posted just two weeks ago, achieves great results and is another step in the same direction as our work.
> > > > > > >
> > > > > > > With that said, we think improving certification as Step 2 is certainly an important line of research as well - it was just not the focus of our work. If the gap in robustness guarantees between verification as Step 2 and certification as Step 2 can be decreased, we feel that is also a valuable research contribution, as certification has the advantage of being faster than verification.
> > > > > > >
> > > > > > > We agree that comparing the runtimes of an exact verifier on networks trained using [9] and our approach can provide further insight, and would be interesting future work.
> > > > > > >
> > > > > > > You also make a great point that exact verification is important in more general settings beyond adversarial robustness. Ultimately, we believe that our techniques can be useful in those other settings of verification as well.
> > > > > > >
> > > > > > > In light of the discussion here regarding certification and verification, we feel that it would improve clarity to add an additional section clarifying these terms and their relation, much like we have tried to do so here, in the Appendix. We also plan to revise our manuscript to cite the very recent work of [9].
> > > > > > >
> > > > > > > Finally, we address potential similarity between our results and https://arxiv.org/pdf/1711.00851.pdf [10] in our next response below.
> > > > > > >
> > > > > > > [9] Gowal, S., Dvijotham, K., Stanforth, R., Bunel, R., Qin, C., Uesato, J., Mann, T. and Kohli, P., 2018. On the Effectiveness of Interval Bound Propagation for Training Verifiably Robust Models. arXiv preprint arXiv:1810.12715.
> > > > > > >
> > > > > > > [10] Eric Wong and J. Zico Kolter. Provable defenses against adversarial examples via the convex outer adversarial polytope. In International Conference on Machine Learning (ICML), 2018.

---

### Official Review · AnonReviewer1 · 2018-11-02
**well motivated and novel method**

**Rating:** 7
**Confidence:** 3

**Review:**

This paper proposes methods to train robust neural networks that can also be verified faster. Specifically, it uses pruning methods to encourage weight sparsity and uses regularization to encourage ReLU stability. Both weight sparsity and ReLU stability reduces time needed for verification. The verified robust accuracy reported in this paper is close to previous SOTA certified robust accuracy, although not beating SOTA.

The paper is clearly written and easy to follow.

The reviewer is familiar with literatures on certifiable robust network literature, but not familiar with verification literature. To the best knowledge of the reviewer, the proposed method is well motivated and novel, and provides a scalable method for verifying (instead of lower bounding) robustness.

Other comments:

I think there should be some discussions on applicability on different robustness measures. The paper focus on L_\infty norm bounded attack, is this method extendable to other norms?

Re: robust accuracy comparison, I found some previous SOTA results missing from Table 3. For example, Mirman et al., 2018 (Appendix Table 6) reached 82% (higher than 80.68% achieved in this paper) provable robust accuracy for MNIST eps=0.3 case. and this is not reported in Table 3. The CIFAR10 results in Mirman et al., 2018 is also better than the best SOTA accuracy in Table 3.


Matthew Mirman, Timon Gehr, and Martin Vechev. Differentiable abstract interpretation for provably robust neural networks. In Jennifer Dy and Andreas Krause, editors, Proceedings of the 35th International Conference on Machine Learning, volume 80 of Proceedings of Machine Learning Research, pages 3575–3583, Stockholmsmssan, Stockholm Sweden, 10–15 Jul 2018. PMLR. URL http://proceedings.mlr.press/v80/mirman18b.html.

---

> ### Author Response · Authors · 2018-11-13
> **Response**
>
> We thank the reviewer for their useful comments. Your comments will help us in revising this paper.
>
> We agree that addressing norms other than L_\infty is an important direction. The techniques explored in our paper are, in general, applicable to other L_p norms (as well as more broader sets of perturbations). Inducing sparsity via L1-regularization and/or weight pruning will still reduce the number of variables in the formulation of verification problems and should improve verification speed. ReLU stability will also help and can still be encouraged via our proposed RS Loss. We do acknowledge that the L_\infty norm will give the tightest bounds on the input layer, which could mean that ReLU stability is easier to optimize for in the L_\infty case.
>
> To clarify with a quick example - if we have a 784-dimensional MNIST input (x1, x2, … x784) with values in the range [0, 1], a reasonable L_\infty norm bound on allowed perturbations may be eps=0.3. On the other hand, a reasonable L_2 norm bound on allowed perturbations may be eps=3. This means that for the L_\infty case, a perturbed input x’ with first dimension x1’ is bounded by x1 - 0.3 < x1’ < x1 + 0.3, while for the L_2 case, the tightest bounds on x1’ are 0 < x1’ < 1. Even though these bounds are looser, encouraging ReLU stability will still improve verification speed.
>
> Finally, as of now, most literature in verification and certification that we are aware of has also focused on the L_\infty norm. Therefore, we similarly chose to focus on it as the most common and natural benchmark. We will be sure to discuss addressing other L_p norms and input constraints in more detail in a revised version of this paper.
>
> Additionally, thank you for pointing out the Mirman et. al 2018 paper - we will absolutely add those relevant results to our comparison tables and references section.

---

### Official Review · AnonReviewer3 · 2018-11-02
**Well motivated, thorough empirical analysis, well written**

**Rating:** 8
**Confidence:** 2

**Review:**

Training for Faster Adversarial Robustness verification via inducing RELU stability


As I am familiar yet not an expert on adversarial training and robustess, my review will focus mainly on the overall soundness of the manuscript. I also only went superficially into the quantitative results.

Summary:

The authors are interested in the problem of verifying neural networks models trained to be robust against adversarial attacks. The focus is on networks with relu activations and adversarial perturbations within an epsilon l1-ball around each input, and the verification problem consists in proving the network performs as intended for all possible perturbations (infinitely many)

The review on verification is clear.
Elements that affect verification time are introduced and well explained in main text or appendix from both intuitive and theoretical perspective: l1 penalty, weight pruning, relu stability. These can be summ\arized as : you want few neurons, and you want them to operate in the same regime for all inputs, both to avoid branching. Relu stability is apparently a new concept and the proposed regularization approximately enforces relu stability.
The approximation [itself using the novel improved interval arithmetic] based bounds on unit activations propagated through the network seems not to scale well with depths (more units are mis-labelled as relu unstable, hence wrongly regularized if I understand correctly). The authors acknowledge and document this fact but I would like to hear more discussion on this feature and on the trade-off that still make this approach worthwhile for deeper networks.

This regularization does not help performance but only paves the way for a faster verification, for this reason the term co-design is used.

The rest of the manuscript is a thorough empirical analysis of the effect of the penalties/regularizations on the network and ultimately on the verification time, keeping an eye on not deteriorating the performance of the network.
How much regularization can be added seems to be indeed an empirical question since networks are ‘over-parametrized in the first place’ with no clear way to a priori quantify task or model complexity.

The devil is in the details and in practice implementation seems not straightforward with a complex optimization with varying learning rates and different regularizations applied at different time along the way. But this seems to be the case for most deep learning paper.

The authors claim and provide evidence to be able to verify network well beyond the scope of what was achievable before due to the obtained speed-ups, which is a notable feature.

Overall, this manuscript is well structured, thorough and pleasant to read, and I recommend it to be accepted for publication at ICLR

---

> ### Author Response · Authors · 2018-11-13
> **Response**
>
> We thank the reviewer for their helpful comments. We are glad you found the paper pleasant to read!
>
> We agree that labeling unstable ReLUs properly is an important aspect of our technique. The upper and lower bounds we compute on each ReLU are conservative - thus, every unstable ReLU will always be correctly labeled as unstable, while stable ReLUs can be labeled as either stable or unstable. Importantly, every unstable ReLU is correctly labeled and penalized by the RS Loss we propose. The tradeoff is that stable ReLUs mislabeled as unstable will also be penalized, which can be an unnecessary regularization of the model.
>
> We showed empirically that we could achieve the following two objectives at once using RS Loss
> 1) Reduce the number of ReLUs labeled as unstable, which is an upper bound on the true number of unstable ReLUs
> 2) Achieve similarly good test set accuracy and PGD-adversarial accuracy as a model trained without RS Loss
>
> For example, when comparing the Control and “+RS” networks for MNIST and eps=0.1, we decreased the average number of ReLUs labeled as unstable (using bounds from Improved Interval Arithmetic) from 290.5 to 105.4 with just a 0.26% loss in test set accuracy (cf. Appendix C.3, Appendix E).
>
> The same trends hold for deeper networks (we only showed results for a 3-layer network in Appendix C.3, but we will include details about a 6-layer network in the revision). For the deeper 6-layer “+RS” network for MNIST and eps=0.1 that we presented, it had a test set accuracy of 98.93% and just 184.6 ReLUs labeled as unstable at the end of training [*]. Training the exact same network without the RS Loss penalty had a slightly higher test set accuracy (99.09%) but also had far more ReLUs labeled as unstable (1028.3). Thus, we could effectively reduce the number of ReLUs labeled as unstable without significantly degrading test set accuracy.
>
> We will clarify these points better in Appendix C where we discuss ReLU bounds when revising the paper.
>
> [*] Edit made: Previously, we wrote test set accuracy of 98.95% and 150.3 ReLUs labeled as unstable, which matches Appendix E. Those are the correct numbers after post-processing (weight pruning and ReLU pruning) is applied, whereas the updated numbers we write here are before post-processing, to match the fact that the 99.09%/1028.3 unstable ReLUs numbers are also computed before post-processing.

---

### Author Response · Authors · 2018-11-26
**Submission Revised**

We would like to thank all reviewers and commenters for their suggestions on improving the manuscript. We have revised our submission based on the feedback we received, and uploaded our revision.

---

### Meta-Review · Area_Chair1 · 2018-12-13
**Borderline accept**

**Confidence:** 4
**Recommendation:** Accept (Poster)

**Metareview:**

This paper introduced a concept called ReLU stability to motivate regularization and enable fast verification. Most of the analysis was presented empirically on two simple datasets and with low-performing models. I feel theoretical analysis and more comprehensive and realistic empirical studies would make the paper stronger. In general, the contribution of this paper is original and interesting.